# Beyond Killing: The Overlooked Contribution of Neutrophils to Tissue Repair

**DOI:** 10.3390/ijms26178669

**Published:** 2025-09-05

**Authors:** Eduardo Anitua, María Troya, Mohammad H. Alkhraisat

**Affiliations:** 1BTI-Biotechnology Institute, 01007 Vitoria, Spain; maria.troya@bti-implant.es (M.T.); mohammad.hamdan@bti-implant.es (M.H.A.); 2University Institute for Regenerative Medicine & Oral Implantology—UIRMI (UPV/EHU-Fundación Eduardo Anitua), 01007 Vitoria, Spain; 3Oral and Maxillofacial Surgery, Oral Medicine and Periodontics Department, Faculty of Dentistry, University of Jordan, Amman 11942, Jordan

**Keywords:** neutrophils, tissue repair, wound healing, platelets

## Abstract

Neutrophils are the most abundant immune cells in humans and the first responders to be recruited at the site of injury. They exhibit high microbicidal activity and a combination of cytotoxic mechanisms that may lead to bystander tissue damage. However, this classical and simplistic view of the neutrophil biology has recently dramatically changed. Emerging evidence indicates an active role for neutrophils in resolution of inflammation and tissue repair. This review specifically explores the mechanisms through which neutrophils perform their anti-inflammatory and tissue-repairing roles, which are also modulated by circadian rhythms—an aspect that influences immune activity and may have implications for treatment timing. A particular focus is placed on the role of platelet-derived products in modulating local neutrophil immune responses. The remarkable phenotypic plasticity of neutrophils and their crucial role in resolving inflammation and restoring homeostasis underscore their promise as a therapeutic approach. However, their activity must be finely regulated to prevent potential tissue damage.

## 1. Introduction

Neutrophils are the most abundant immune cells in humans, representing about 50–70% of the circulating leukocytes. They are formed withing the bone marrow from myeloid precursors in a process called granulopoiesis. This differentiation process is controlled by several cytokines, among which mainly granulocyte colony-stimulating factor (G-CSF) is found [1,2,3]. Neutrophils are considered short-lived cells; nevertheless, following tissue damage, their life span increases as antiapoptotic genes are transiently upregulated [4,5,6]. Neutrophils are the first responders to be recruited at the site of the injury via the release of an array of signals [2,3,5,7]. They exhibit a high microbicidal activity with an extraordinary array of receptors for the recognition of pathogens and a combination of cytotoxic mechanisms, including production of reactive oxygen species (ROS), neutrophil extracellular traps (NETs), inflammatory mediators, and proteolytic enzymes [6,7,8]. Traditionally, these neutrophil functions were thought to lead to bystander tissue damage [2,9]. More recently, advanced techniques, such as intravital microscopy, genetic fate mapping, transgenic animals, and single-cell sequencing, have drastically changed our simplistic view of the neutrophil biology [2,3,10]. This emerging evidence indicates an important role for neutrophils in resolution inflammation and tissue repair through multiple mechanisms involving anti-inflammatory, pro-angiogenic, and regenerative neutrophil functions [7,11]. Beyond their established role in hemostasis and thrombosis, platelets are increasingly recognized as key regulators of the inflammatory and immune response [12,13,14]. The crosstalk between platelets and innate immune cells promotes thrombosis, inflammation, and tissue damage but also regulate the resolution of inflammation, tissue repair, and wound healing [14]. Based on this premise, platelet-derived components may contribute significantly to the immune regulation of neutrophil function.

In this state-of-the-art review, we present a conceptual framework that positions neutrophils as dynamic regulators of tissue homeostasis and repair. Traditionally viewed as mere frontline defenders, neutrophils are now emerging as central orchestrators of resolution and regeneration. We highlight their evolving roles in tissue repair processes and examine their functional interplay with platelets, particularly in the context of platelet-rich plasma (PRP) therapies. The review integrates emerging insights under a unified paradigm of immune modulation and tissue restoration.

## 2. Neutrophils and Tissue Repair

Neutrophils, as part of the innate immune system, are classically considered the first line of defense against infection and tissue injury in response to a plethora of inflammatory cues [5,11,15]. Neutrophil recruitment is driven by the release of damage-associated molecular patterns (DAMPs) from injured cells or pathogen-associated molecular patterns (PAMPs) during infection, which activates inflammatory responses through pattern recognition receptors (PRRs) [5,16]. The release of neutrophils follows a chemokine-regulated, circadian fashion [15]. Following recruitment, neutrophil activation and degranulation occurs, releasing potentially cytotoxic content [16,17]. Nevertheless, emerging evidence indicates that neutrophils also play an active role in the inflammation resolution process, thus contributing to tissue repair via multiple mechanisms that can occur simultaneously or sequentially [2,17] (Figure 1).

### 2.1. Clearance of Debris

As professional phagocytes, neutrophils are involved in clearing necrotic cellular debris at the site of injury. Necrotic cell debris contains several DAMPs (DNA, histones, adenosine triphosphate (ATP), high mobility group box 1 (HMGB1), and actin, among others) that interact with PRRs, triggering excessive inflammation and potentially leading to autoimmunity [18,19,20]. Therefore, depletion of neutrophils or inhibition of their function may impair or substantially delay tissue regeneration as the removal of necrotic cell debris is critical for achieving tissue healing and recovering homeostatic conditions [9,21] (Figure 2).

Neutrophils have been reported to differentially express DNA-degrading and repair-associated genes and proteins following acid-induced lung injury, which favors optimal organ repair. In fact, intra-tracheal DNAse I administration enhanced alveolar repair in neutropenic mice with an associated increase in DNA-containing debris. This neutrophil-dependent DNA degradation occurred in a MyD88-dependent pathway [22]. Similarly, in a model of sterile liver injury, DNA clearance was also necessary to promote revascularization as neutrophil depletion resulted in an increase in remnants in the injury site [23]. Complement activation also contributes to the removal of necrotic cell debris by neutrophils, as reported by Vandendriessche et al. [19]. They showed that neutrophils relied on complement proteins to phagocytose debris in both chemical and thermal liver injury models. In fact, the absence of C1q and C3 led to a significant accumulation of necrotic debris. Human neutrophils exhibited a pro-resolving phenotype after the clearance of opsonized necrotic debris. Clearance of myelin debris, which contains many inflammatory, growth inhibitory, and neurotoxic factors, is an essential process to limit tissue injury and promote functional healing in central nervous system (CNS) disorders [24,25]. Wallerian degeneration (WD) involves axonal degeneration, myelin clearance, and changes in the extracellular matrix, all of which contribute to the restoration of nerve integrity [26]. Neutrophils have been reported to play an essential role during WD. In fact, Lindborg et al. [27] showed that neutrophils can replace CCR2^+^ macrophages as the primary phagocytic cells involved in the clearance of axonal and myelin debris after nerve injury, as evidenced by the inhibition of myelin clearance after neutrophil depletion in both wild-type and Ccr2^−/−^ mice. Similarly, the neutrophil peptide 1, an immunoregulatory polypeptide secreted by neutrophils, has been described to accelerate WD by activating macrophages [28,29]. Neutrophils could also favor muscle regeneration by removing remnants from the injured area. Extensive necrotic debris was observed in mice treated with antigranulocyte monoclonal antibodies after myonecrosis induced by *Bothrops asper* venom and myotoxin I [30], indicating that macrophages alone cannot perform this function. In the setting of acute myocardial infarction (MI) [31], the secretion of neutrophil gelatinase-associated lipocalin (NGAL) favors cardiac repair and function by leading to macrophage phenotypic shift towards a pro-resolving one and promoting efficient debris removal. This also confirmed the vital role of the neutrophil secretome in the clearance of debris for the healing response.

### 2.2. Pro-Resolving Mediators

Neutrophils release numerous mediators, which directly contributes to the end of inflammation and restoration of tissue homeostasis and functionality [2,18,32,33] (Figure 2). Early angiogenesis plays a crucial role in restoring the proper tissue structure and function [34]. In fact, impaired tissue vascularization is linked to chronic inflammation. Polymorphonuclear (PMN) leukocytes are heavily involved in both physiological and tissue injury-derived angiogenesis. In fact, during the menstrual cycle, neutrophils secrete vascular endothelial growth factor-A (VEGF-A) to promote angiogenesis in the endometrial tissue [35]. These immune cells are a highly plastic and heterogeneous population. A distinct proangiogenic neutrophil population, which exhibits high expression of the VEGF-A receptor VEGFR1, chemokine receptor CXCR4, and integrin alpha 4 (CD49d), was recently identified in humans and mice [2,36]. Inhibiting recruitment of VEGFR1^+^ resulted in an impaired angiogenesis in ischemic tissues [37,38]. Matrix turnover is also crucial to tissue repair. Matrix metalloproteinases (MMPs) are a group of proteolytic enzymes that play a crucial role in matrix degradation, enabling the endothelium to sprout and reorganize to form new blood vessels, and facilitating the release of various growth factors bound to the extracellular matrix (ECM) [18,39,40,41]. The specific subpopulation of angiogenic neutrophils secretes MMP-9, which, in turn, promotes the release of proangiogenic matrix-bound growth factors, such as VEGF and fibroblast growth factor-2 (FGF-2) [36,42]. Neutrophil-derived MMP-9 is stored in tertiary granules, which are the first to be degranulated with the lowest levels of stimulation [41]. Neutrophils show a unique feature as they are capable of releasing MMP-9 free of its endogenous tissue inhibitor of metalloproteinases-1 (TIMP-1) and thus make it readily available for activation and further enzymatic functions. This was confirmed by natural and stoichiometrically generated and purified proMMP-9/TIMP-1 complexes, which failed to stimulate angiogenesis. This finding may underlie the high angiogenic potential of neutrophils [39]. Neutrophils contribute to tissue repair through mechanisms involving MMP-9. Cells in pancreatic islet secrete high amounts of the most potent angiogenic factor, VEGF-A. In an avascular transplanted mouse pancreatic islet, Christofferson et al. [43] found that VEGF-A, released during hypoxia, recruited a CD11b^+^/Gr-1^+^/CXCR4^hi^ proangiogenic neutrophil subset, which in turn contained more MMP-9 required for islet revascularization and restoration of perfusion. Metalloproteinases have been also implicated in ischemic tissue regeneration, which greatly depends on neovascularization. Activation of MMP-2 and MMP-9 is neutrophil-dependent in limb ischemia and might be necessary for tissue remodeling and angiogenesis to occur [44]. In addition, G-CSF stimulates neutrophil-driven angiogenesis in an MMP-9-dependent fashion. The release of VEGF from G-CSF-stimulated neutrophils is impaired in MMP-9-deficient ischemic mice [38]. MMP-9 activation is also required for VEGF-induced tube formation [45]. In myocardial tissue repair, reperfusion is essential. Canine neutrophil-derived MMP-9 was also reported to be involved in this process, thus accelerating healing [41]. Angiogenesis and osteogenesis are highly interconnected biological processes in bone tissue repair. In a triple human cell co-culture model consisting of osteoblasts, endothelial cells, and neutrophils, inclusion of the latter increased the formation of microvessel-like structures and significantly induced the expression of proangiogenic markers such as VEGF-A, CD34, EGF, and FGF-2 in a dose- and time-dependent manner. MMPs were also upregulated by the inclusion of neutrophils in the co-culture model. In addition to stimulating angiogenesis, the inclusion of neutrophils enhanced major osteogenic markers [42]. Neutrophil-derived MMPs have also been described as mediators of liver repair. Neutrophil-derived MMP-8 or neutrophil-mediated activation of intrinsic hepatic MMPs was responsible for successful repair in a murine model of cholestatic injury and repair [46]. Inhibition of neutrophil-derived MMPs impaired the resolution of the pathogenesis of liver injury induced by acetaminophen (APAP) administration [47].

Neutrophils also possess the significant capacity to generate pro-resolving molecules, including annexin A1 and specialized pro-resolving mediators (SPMs), which regulate the resolution of tissue inflammation [48,49,50]. SPMs are lipid mediators derived from polyunsaturated fatty acids, such as AA (arachidonic acid), EPA (eicosapentaenoic acid), DHA (docosahexaenic acid), and n-3 DPA (n-3 docosapentaenoic acid). They include resolvin, protectin, maresins, and lipoxins (LXs) [50,51,52]. The resolution of inflammation is an active, time-dependent process involving a biosynthetic transition from pro-inflammatory to pro-resolving mediators. The infiltrating neutrophils switch from producing pro-inflammatory mediators (leukotrienes and prostaglandins, plus AA-derived products) to SPMs [32,51,53]. They inhibit neutrophil recruitment, decrease vascular permeability, induce neutrophil apoptosis, and promote their efferocytosis [2,32]. SPMs promote their biological effects by acting on G protein-coupled receptors (GPCRs), which, along with SPMs, have been proposed as therapeutic targets [51,54]. A large body of experimental evidence suggests that exogenous application of SPMs could effectively enhance the resolution of inflammation and promote tissue repair. This issue has been described in detail elsewhere [51,53]. However, in this section, we emphasize the molecules endogenously released by neutrophils and their potential reparative effects. In this regard, single-cell RNA sequencing analysis has identified a population of resolving neutrophils in murine neutrophils, characterized by, among other markers, elevated expression of Resolvin D1. This subpopulation increased following treatment with sodium 4-phenylbutyrate (4-PBA), an aromatic fatty acid analogue with therapeutic potential. The generation of these resolving neutrophils involves the PPARγ/LMO4/STAT3 signaling pathway [55]. Lipid mediator profiling of inoculated mice with *E. coli* demonstrated that N-3 docosapentaenoic acid-derived resolvin D5 was produced during bacterial infections and that its presence was guaranteed during the resolution phase of inflammation [56]. In addition, hypoxic environments, physiologically present in some cellular niches and in inflammation, activate SPM biosynthesis. Norris et al. [57] identified a novel resolvin (RvE4). In fact, modifying the metabolism of neutrophils by inhibiting glycolysis increased their production under these hypoxic conditions, ultimately enhancing the resolution of inflammation. Dalli et al. [58] also reported anti-inflammatory effects of annexin 1 in murine PMN-derived microparticles. Recent discoveries have identified new factors influencing SPM formation in human neutrophils. The biosynthesis of E- and D-series resolvins is linked to intact cells, while lipoxin formation occurs independently of membrane integrity. In fact, lipoxin production is significantly upregulated when cellular integrity is compromised [59].

Moreover, other molecules also contribute to the modulatory role of neutrophils. Although typically associated with harm, ROS also exhibit dual biological effects. In a mouse model of APAP-induced liver injury, neutrophils facilitate optimal liver repair by promoting the conversion of macrophages to a pro-resolving phenotype through the production of ROS [60]. The NF-κB pathway plays a crucial role in regulating immune and inflammatory responses. Human neutrophils can suppress the inflammatory signaling in macrophages, independently of neutrophil viability, by inhibiting macrophage NF-κB activation through the blockade of the TAK1-IKKβ axis [61]. In response to infection, neutrophils have been reported to produce the anti-inflammatory cytokine interleukin-10 (IL-10), both in humans and mice [62,63,64,65]. Indeed, as will be further elucidated in subsequent sections, this molecule plays a significant role in multiple processes associated with neutrophil-mediated tissue repair, as it is indirectly involved in both efferocytosis and the mechanism of action of neutrophil extracellular traps (NETs).

### 2.3. Efferocytosis

Neutrophils are essential in controlling inflammation by modulating macrophage function through efferocytosis (Figure 2). This process involves the phagocytosis of apoptotic cells, which is crucial for maintaining tissue homeostasis and resolving inflammation. Engulfment of apoptotic neutrophils by macrophages triggers anti-inflammatory signaling, prevents neutrophil lysis, and reduces immune responses [3,9,66]. Efferocytosis prompts macrophages to adopt M2 phenotypes, characterized by reduced pro-inflammatory cytokines and increased anti-inflammatory and reparative cytokines such as transforming growth factor-β1 (TGF-β1) and IL-10 [67,68]. Efferocytosis also stimulates the synthesis of lipoxins, docosahexaenoic acid (DHA) products, and resolvins while reducing the production of classical eicosanoids, thereby facilitating the tissue repair phases [69,70,71,72,73]. Indeed, impaired efferocytosis is associated with the progression of various chronic inflammatory and autoimmune diseases, including chronic obstructive pulmonary disease, asthma, idiopathic pulmonary fibrosis, rheumatoid arthritis, systemic lupus erythematosus, atherosclerosis, type I diabetes, and cardiovascular diseases [68,74,75].

Phagocytes interact with apoptotic cells through a specific set of markers known as “apoptotic cell-associated molecular patterns” (ACAMPs), which are recognized by distinct receptors and bridging molecules [76]. Apoptotic neutrophils generate “find me” and “eat me” signals to promote efferocytosis [67,68,70,76,77,78]. Recruitment of phagocytes to apoptotic cells is driven by the release of chemotactic “find me” signals, including lysophosphatidylcholine, CX3CL1 (fractalkine), nucleotides ATP and uridine 5′ triphosphate (UTP), and sphin-gosine 1-phosphate [70,76]. Recognition of apoptotic cells by phagocytes mainly occurs via phosphatidylserine (PS), the most universal and best characterized “eat me” signal, which is normally localized in the inner leaflet of the plasma membrane of viable cells and is externalized during apoptosis [6,68,69,75,77,79]. Additionally, apoptotic neutrophils upregulate annexin-I and calreticulin on their surface, which act as supplementary “eat me” signals [67,69,70,79,80]. “Eat me” signaling is counter-balanced by expression levels of “don’t eat me” molecules such as CD47, CD31, and plasminogen activator inhibitor-1 (PAI-1) [67,76]. Many “eat me” signals are directly recognized by phagocyte receptors, including members of the T-cell immunoglobulin domain and mucin domain (TIM) family, complement receptors CR3 and CR4, scavenger receptors SRA and CD36, mannose receptor (MR), BAI1, and integrins α5β3 and α5β5 [67,68,69,70,76,81]. Other signals, like MERTK and complement receptors, require bridging molecules such as growth arrest specific gene 6 (GAS6), protein S, complement C1q, thrombospondin, and milk fat globule protein (MFG-8) to interact with Tyro/Axl/Mer (TAM) family macrophage receptors for their recognition [68,69,70,81,82]. Following these signals, the cytoskeleton must undergo rearrangement, enabling the internalization of apoptotic cells. Rac1, a member of the Rho family GTPases, is a key molecule involved in the internalization of apoptotic cells during efferocytosis [78,79]. Its activation triggers actin polymerization, an essential process for the engulfment of apoptotic cells. Finally, the processing of engulfed cellular cargo requires a non-canonical LC3-associated phagocytosis (LAP) that occurs within the phagolysosomal compartment in an immunologically silent fashion, a hallmark of efferocytosis compared to phagocytosis of pathogens [69,70,78,79]. Macrophages adjust their energy metabolism to maintain appropriate levels of intracellular content by regulating lipid metabolism, among others. They activate lipid-sensing nuclear receptors (PPAR and LXR), which also contribute to the anti-inflammatory response [70,78,83].

Over the past years, an increasing number of biomedical research studies have been emerging to elucidate the role of efferocytosis in physiological and pathological contexts, exploring the mechanisms that underlie these processes and its implications for cellular homeostasis and disease progression. In lung injuries, neutrophil efferocytosis switches alveolar macrophages to prioritize the resolution of inflammation over antibacterial responses through the inhibition of mitochondrial ROS by the neutrophil myeloperoxidase (MPO), which, in turn, enhances uncoupling protein 2 (UCP2) expression [84]. This UCP2 upregulation further enhances efferocytosis, supporting the resolution of lung inflammation. While this effect of efferocytosis on mitochondrial metabolism is typically beneficial, fostering tissue repair, it also creates a vulnerability to secondary bacterial infections during the resolution phase of inflammation. PPARγ activation following apoptotic cell instillation in a bleomycin-induced lung injury in mice promoted the resolution of lung inflammation and fibrosis through the regulation of alveolar macrophages’ efferocytosis and pro-resolving cytokines (TGF-β, IL-10, and HGF) [85]. Data from mouse influenza infection models [86] showed highly motile apoptotic neutrophils during the resolution phase, with an active migration pattern of neutrophils towards phagocytes during early apoptosis. The disappearance of neutrophils was predominantly driven by efferocytosis. The secretome from an anti-inflammatory subpopulation of human neutrophils (N2) reprogrammed human macrophages towards a healing phenotype with increased efferocytosis capacity [82]. The authors showed that factors released by N2 neutrophils promoted the upregulation of anti-inflammatory markers, including CD206, TGF-β, and IL-10, as well as nuclear factors linked to a reparative phenotype, such as PPAR, Nur77, and KLF4, in macrophages. Efferocytosis receptors (MerTK, CD36, CX3CR1, and integrins v/5) and bridging molecules (Mfage8 and Gas6) were also induced in macrophages by the anti-inflammatory neutrophil subpopulation. High-mobility group box 1 (HMGB1), a nuclear nonhistone DNA-binding alarmin involved in inflammation, has been shown to inhibit efferocytosis [87,88,89,90,91,92]. The mechanisms responsible for HMGB1-mediated inhibition of efferocytosis are varied: it can bind to phosphatidylserine [88], associate with Src kinase to disrupt its interaction with FAK [89], bind to the α_v_β_3_-integrin [91], or suppress the Rab43-regulated anterograde transport of CD91 [92]. In addition, Banerjee et al. [90] demonstrated that the HMGB1 C-terminal acidic tail was responsible for the inhibitory effects of HMGB1 on efferocytosis. On the contrary, activation of AMP-activated protein kinase (AMPK) was reported for enhancing efferocytosis activity [87,93]. Additionally, the cellular communication network factor 1 (CCN1), a matricellular protein, also induces macrophage efferocytosis of apoptotic neutrophils through the engagement of integrin α_v_β_3_ [68,94].

Therefore, neutrophils are typically cleared through a silent process involving apoptosis, followed by their engulfment by phagocytic scavengers. However, when the rate of apoptosis exceeds the scavenging capacity or when efferocytosis directly fails, apoptotic cells progress to secondary necrosis. Unlike efferocytosis, secondary necrosis is a pro-inflammatory, immunogenic, and autolytic event characterized by disruption of the cell membrane and the extracellular release of cellular content [77,95,96]. Neutrophil cannibalism can also occur when efferocytosis fails as a result of extensive neutrophil infiltration. In such cases, neutrophils actively contribute to clearance by engulfing apoptotic cells and/or debris resulting from secondary necrosis. In a mouse model of lipopolysaccharide-induced lung inflammation [95], electron microscopy revealed that nearly 50% of neutrophils in the affected tissue contained phagosomes filled with material from other neutrophils. Neutrophil cannibalism is actively involved in the resolution of inflammation [79]. This process prompts neutrophils to release TGF-β1, which subsequently regulated both IL-1β production and the respiratory burst in a mouse model of monosodium urate monohydrate crystal-induced inflammation and gout [97].

### 2.4. Neutrophil Extracellular Traps (NETs)

Neutrophil extracellular traps (NETs) were first described in 2004 by Brinkmann et al. [98]. NETs are web-like structures of decondensed chromatin coated with citrullinated histones and granule-derived proteins such as neutrophil elastase (NE), cathepsin G, and MPO [70,99,100,101]. NETs are released in response to a range of both infectious and sterile stimuli, including bacteria, cytokines, immune complexes, uric acid crystals, phorbol esthers, DAMPs, and calcium ionophores [70,102,103]. NETs play a crucial role in host defense by trapping pathogens and acting as a physical barrier to prevent the spread of pathogens and inflammatory mediators, thereby protecting larger areas of tissue from damage [100,101,103,104]. Additionally, NETs can degrade inflammatory mediators to help resolve inflammation driven by neutrophils [104]. However, depending on the microenvironment, NETs can also exhibit immunogenic and pro-inflammatory properties [70,105]. Therefore, the formation and removal of NETs should be delicately balanced. The term “NETosis” has been used for long time to describe the process leading NET formation. However, this process does not always result in the death of neutrophils. For that reason, in 2018, the Nomenclature Committee on Cell Death [106] suggested avoiding the term “NETosis” unless there was clear evidence of cell death. Some authors recommend the use of the term “NET formation” instead [99,100,107]. Different types of NET formation have been described: suicidal, vital, and mitochondrial [103,107,108,109,110,111] (Figure 2). The classic suicidal NET formation involves the final neutrophil lysis and is characterized by nuclear chromatin decondensation, nuclear membrane rupture, and NET release. This process is dependent on NADPH oxidase-2 (NOX) and usually takes 2–4 h. In contrast, vital NET formation preserves neutrophil viability and functionality after releasing NETs, occurring within 30–60 min, and follows a NOX-independent pathway. In both types, peptidylarginine deiminase 4 (PAD4) mediates histone citrullination, facilitating chromatin decondensation, with NE and MPO also playing roles. Finally, mitochondrial NET formation represents a distinct pathway that triggers NET formation from mitochondrial DNA instead of nuclear DNA.

As with almost everything related to neutrophils, the release of NETs is a double-edge sword, and so maintaining a balance between formation and clearance of NETs is necessary [112,113]. At high neutrophil densities, NETs exhibit a tendency to aggregate (aggNETs) and degrade pro-inflammatory mediators via serine proteases promoting the resolution of inflammation, thus contributing to wound healing [100,104,113,114,115]. Schauer et al. [114] demonstrated that, under high neutrophil densities, NET formation and the aggregation of NETs promoted the resolution of monosodium urate (MSU) crystal-induced inflammation in gout by degrading neutrophil-derived inflammatory mediators via serine proteases. Furthermore, they described the same mechanism to mitigate neutrophilic inflammation in a zymosan-induced arthritis model using yeast cell-wall polysaccharide. According to Hahn et al. [116], the proteolysis of inflammatory mediators by NETs is also largely dependent on neutrophil density, requiring at least intermediate cell densities. Additionally, this proteolytic degradation was absent in Papillon–Lefèvre syndrome neutrophils, which are specifically characterized by nonfunctional neutrophil serine proteases and are associated with severe periodontal inflammation. The resolving role of aggNETs was also mediated through histone degradation, as extracellular histones are reported to be cytotoxic [117,118]. Knopf et al. [102] showed that aggNETs sequestered and degraded histones using two serine proteases, NE and proteinase 3, resulting in a reduced cytotoxic effect on epithelial cells. In another in vitro study [119], low NET concentrations also positively affected human keratinocyte behavior by increasing cell proliferation and improving wound closure. This was mediated by activation of the NFκB signaling pathway. AggNETs also provide a physiologically protective and resolving role on the ocular surface [115]. After overnight sleep or after prolonged eye closure, continuous neutrophil infiltration occurs, which leads to the formation of eye rheum, a form of aggNET. Due to the proteolytic activity of eye rheum, the ocular surface remains clean and free from inflammation. AggNETs was also reported to be essential for preventing chronic inflammation, as tissue-damage-induced inflammation caused by nanodiamond injections failed to resolve in Ncf1 mice deficient in NAPDH-dependent ROS [113]. This pro-resolutive role of NETs has been also described in other pathological conditions. Neutrophils utilize PAD4 to transform blood clots into immunothrombi on mucosal ulcerations rich in NETs by decreasing serpin activity, thus aiding wound healing, and preventing ulcer bleeding in severe human inflammatory bowel disease and an experimental model. Therefore, NET-associated immunothrombi offer protection in acute colitis by controlling rectal bleeding [120]. Type 2 diabetes mellitus (T2D) is associated with an increased risk of bacterial infections and delayed wound healing. Neutrophils of both hyperglycemic and normoglycemic T2D patients released NETs decorated with the cathelicidin antimicrobial peptide LL-37, but without antibacterial function. Treatment with macrolide antibiotic clarithromycin increased externalization of LL-37 on NETs from well-controlled T2D neutrophils, restoring their antibacterial activity, and thus improving the wound-healing capacity of fibroblasts [121]. Ribon et al. [122] conducted a large analysis comparing rheumatoid arthritis (RA) and healthy donor (HD) NETs on RA and HD target cells. NETs were reported to exhibit dual activity, being both pro- and anti-inflammatory depending on the context. They inhibited IL-6 secretion by lipopolysaccharide-activated macrophages, and the co-factors C1q and LL-37 reinforced this inhibition. IL-10 secretion was also stimulated. Furthermore, AggNETs act as a transient barrier to limit the spreading of necrotic tissue in patients with necrotizing pancreatitis and peritonitis [123]. On the other hand, Manda-Handzlik et al. [112] reported that only the secretome of M1 and M2 macrophages could diminish NET formation, while other immune cells or cytokines were ineffective. In fact, the clearance of NETs is primarily achieved through the action of extracellular DNases and the degradation by macrophages [100,124,125].

### 2.5. Neutrophil Reverse Migration

It was traditionally believed that the resolution of neutrophil-mediated inflammation occurred through the apoptosis of neutrophils and their subsequent clearance by macrophages [7,126,127,128]. However, neutrophils do not always die at the site of injury and instead they leave the source of inflammation, as early evidenced Hughes et al. [129] when tracking radiolabeled neutrophils in a model of glomerular capillary injury rats. More recently, this process was directly visualized in vivo for the first time in zebrafish larvae using time-lapse imaging [130]. This novel mechanism introduced the concept of “neutrophil reverse migration” as the process whereby neutrophils migrate away from the source of inflammation [131,132,133] (Figure 2). The exact functional role of this process is yet to be fully understood, as evidence suggests it can have both protective and harmful effects depending on the context. Neutrophil reverse migration can promote wound healing by improving the clearance of neutrophils but may also contribute to the spread of systematic inflammation [128,134,135]. Within the broader term of reverse migration, different types have been encompassed: reverse migration, reverse interstitial migration, reverse luminal crawling, or reverse transendothelial cell migration [134]. Reverse-migrated neutrophils have a distinct surface phenotype [136]. This subset of human neutrophils was found to be CD45-high and CXCR1-low, which represents a different profile from naïve, circulating, and tissue-infiltrating neutrophils [136]. Overall, the mechanisms driving reverse migration are still not clear. However, factors such as neutrophil–endothelial interactions, lipid mediators, and chemokines along with their receptors have been recognized as contributors to neutrophil reverse migration [135]. The junctional adhesion molecule (JAM) pathway was reported to be involved in neutrophil reverse migration in mouse models. Colom et al. [137] found that leukotriene B_4_ (LTB4) stimulated NE to hydrolyze endothelial JAM-C protein, significantly promoting neutrophil reverse migration. Aligned with these finding, Li et al. [138] reported that blockade of the LTB4-specific receptor BLT1 reduced neutrophil reverse transendothelial cell migration (rTEM), and blockade of JAM-C reduced neutrophil reverse migration in septic mice [139]. Therefore, lipid mediators have also been recognized as contributors to neutrophil reverse migration. Findings from an in vivo zebrafish model of inflammation demonstrated that prostaglandin E2 (PGE2) stimulated lipoxin A4 (LXA4) production via the EP4 receptor, contributing to inflammation resolution through reverse migration [140]. According to this, Hamza et al. [141], using a microfluidic platform, also reported an enhancement of reverse migration in the presence of LXA4. The emergence of microfluidics has provided an enabling ex vivo approach to understand the interaction and behavior of neutrophils [142]. In this context, Wang et al. [143] developed a microfluidic device that enabled the identification of four distinct neutrophil migration patterns. The authors demonstrated that reverse migration was more abundant in the IL-8 gradient than in the LTB4 and fMLP chemokine gradients. Regarding neutrophil movement, Holmes et al. [132] provided evidence that neutrophil reverse migration is a random process rather than fugetaxis. Several studies have also identified different chemokines to be involved in neutrophil reverse migration. Isles et al. [144] reported that inhibition of the CXCL12/CXCR4 signaling axis increased neutrophil reverse migration in a transgenic zebrafish model. Wang et al. [23] showed that neutrophils migrated via CXCR4 to the bone marrow in sterile thermal hepatic injury in mice. Furthermore, the inhibition of CXCR1/2 was found to prevent neutrophil reverse migration, thereby hindering zebrafish heart regeneration following cryoinjury [145]. Other molecules have also been described as playing a significant role in this process. In fact, activation of hypoxia-inducible factor-1 (HIF-1) also reduced both neutrophil apoptosis and reverse migration after tail transection in zebrafish larvae, delaying inflammation resolution [146], and IRG1/ACOD1 promoted neutrophil reverse migration in a mouse model of lipopolysaccharide-induced acute lung inflammation [147]. Macrophages also contribute to neutrophil reverse migration, even though direct interaction between them is not a necessary condition for the process to occur [140]. Macrophage redox-Src family kinase signaling was reported to be necessary for neutrophil reverse migration [148], as well as anti-inflammatory macrophage-derived extracellular vesicles in sterile injury [149].

### 2.6. Circadian Rhythm

Circadian rhythms regulate a wide range of physiological processes, including immune responses, and exhibit a 24 h oscillatory cycle synchronized with the Earth’s rotation. This synchronization enables organisms to anticipate predictable environmental cues, such as sunlight and food availability [150,151,152,153,154]. Mammalian circadian rhythms arise from a hierarchical system of biological clocks, with a master clock in the suprachiasmatic nuclei (SCNs) of the brain coordinating peripheral clocks distributed across multiple tissues and organs via neural and humoral mediators [153,155,156,157]. The circadian clock operates through two main transcription–translation feedback loops. The central loop involves a heterodimer composed of the transcription products of BMAL1 (brain and muscle ARNT-like 1) and CLOCK (circadian locomotor output cycles kaput). This complex binds to E-box sites, promoting the expression of inhibitory factors PER and CRY. As the levels of these inhibitory factors decline, inhibition weakens, initiating a new cycle [152,155,156,157]. An auxiliary loop composed of the transcription factors retinoic acid-related orphan receptor (ROR) and reverse-erb (REV-ERB) also regulates the expression of BMAL1 [155,157]. At the molecular level, most immune cells, including neutrophils, autonomously express these biological clock-regulating genes [153,155,158,159]. Neutrophil circadian trafficking is also rhythmically orchestrating [152,160,161]. The migration of neutrophils is regulated by the Bmal1/CXCL2/CXCR2 axis (Figure 3). Fresh neutrophils released from the bone marrow undergo BMAL1-dependent aging. BMAL1 directly controls the expression of the chemokine CXCL2, which binds to the receptor CXCR2, initiating neutrophil aging and promoting their egress. Conversely, CXCR4 signaling, through its ligand CXCL12 (the bone marrow stromal cell chemokine), inhibits this aging process and favors the retention of neutrophils in the bone marrow. Ultimately, aged neutrophils are cleared from circulation [158,160,161,162,163]. The impact of circadian regulation on circulating neutrophils is evident in infection and vascular inflammation. Research increasingly shows that neutrophil responsiveness fluctuates depending on the time of day, highlighting its dynamic role in immune function [156,161]. Chen et al. [151] found that neutrophil recruitment in a zebrafish model was rhythmically regulated by the Clock 1a protein, with ROS involved in this process. The phagocytic activity of human and murine neutrophils was also reported to follow circadian dynamics [156,164]. Circadian oscillations in neutrophil behavior also altered the outcomes of kidney injury caused by cholesterol crystal injection [154]. The distribution of young and aged cells within the peripheral neutrophil pool was found to display a daily rhythm, and hence neutrophil reactivity, with a relative higher proportion of human aged neutrophils reported in the evening [156]. The neutrophil proteome in both mice and humans undergoes circadian oscillations, with the granule protein content and NET-forming capacity diminishing as neutrophils age in circulation and migrate into tissues [165]. Researchers also observed that this daily variability was abolished in neutrophils deficient in Bmal1, CXCR2, or CXCL2. Additionally, a retrospective analysis revealed a higher incidence and severity of lung disease in the morning, coinciding with an increased presence of young-like neutrophils characterized by a greater granule content and NET-forming capacity. This suggests that circadian modulation of the neutrophil proteome may enhance disease outcomes [165]. The circadian dynamic of neutrophils may also affect stroke outcomes [166]. Metabolic functions are strongly interconnected with immune functions and also exhibit circadian rhythms [167]. Neutrophil-specific *Bmal1* deficiency led to a different anti-inflammatory phenotype, with aging features, under a chronic high-fat diet [163]. Neutrophil *Bmal1* knockout (NE-BKO) mice displayed an increased presence of M2 macrophages and elevated expression of annexin A1 and Kruppel-like factor 4, both of which drive M2 polarization. As a result, *Bmal1*-deficient neutrophils attenuated adipose inflammation induced by a high-fat diet. Epidemiological studies also indicate that acute myocardial infarction (MI) occurs more frequently in the morning and is linked to worse outcomes [168,169]. One potential mechanism underlying this trend involves rhythmic oscillations in cardiac neutrophil recruitment, which are dependent on CXCR2 [168].

This circadian regulation of neutrophils could contribute to personalized medicine in clinical settings by tailoring treatments to temporal variations. Integrating chronotherapy may improve therapeutic efficacy and optimize patient outcomes [158,161,166].

### 2.7. Neutrophil Extracellular Vesicles

Extracellular vesicles (EVs) are nanosized particles released by cells and delimited by a lipid bilayer with no replicate capacity, which play an important role in intercellular communication [170,171]. There are various subtypes of EVs. Neutrophils predominantly release medium-sized vesicles (traditionally referred to as ectosomes) [172,173,174] (Figure 2). Neutrophil EVs (nEVs) are released both during homeostasis and upon activation in response to different stimuli, including chemokines, cytokines, or bacteria and their byproducts [70,171,173]. An increase in intracellular calcium concentration, cytoskeletal rearrangement, and cell membrane budding are crucial during nEV biogenesis [58,172]. They contain cytosolic F-actin and bind Annexin V, indicating that phosphatidylserine (PS) is exposed. nEVs also express a subset of cell surface proteins, including CD11b, CD66b, CD18, and L-selectin (CD62L). Moreover, they include a range of intracellular proteins, such as MMP-9, MPO, elastase, and proteinase 3 [58,173,175]. They carry out their biological functions by delivering a cargo of nucleic acids, proteins, lipids, and metabolites, which are influenced by the cell of origin and its activation state [170,171,172,176].

nEVs show antibacterial effects and anti-inflammatory benefits by reprogramming macrophages, lowering pro-inflammatory cytokines, and enhancing pro-resolving cytokines and lipid mediators in different scenarios. In the context of experimental inflammatory arthritis, nEVs have been reported to lead to cartilage protection. Thomas et al. [177] identified microRNA (miR)-455-3p as one responsible for neutrophil EV chondroprotection. The pro-resolving protein annexin A1 (AnxA1) was also described as a potential effector of EV in arthritis. nEVs, enriched in AnxA1, induced TGF-β1 production, ECM deposition, and chondrocyte protection against apoptosis [178]. Moreover, AnxA1 contributed to the anti-inflammatory properties of nEVs by decreasing neutrophil recruitment and preventing neutrophil–endothelial cell interactions [58]. nEVs also promoted bone regeneration by increasing key osteogenic proteins SOD2 and GJA1 in bone marrow mesenchymal stem cells (BMSCs) [171]. nEVs played a role in allergic contact dermatitis by modulating monocyte-derived dendritic cell maturation in response to nickel, a common chemical sensitizer. This effect was mediated by the downregulation of p38MAPK and NF-κB, resulting in the reduction in pro-inflammatory cytokines [176]. The macrophage response was also reported to be down-modulated by nEVs through several mechanisms: promoting the release of TGF-β1 [179,180,181], inducing an immediate Ca^2+^ flux [180], activating the MerTK [180,182] and PI3K/Akt pathways, and inhibiting the NFκB pathway [182]. This immunomodulatory ability of nEVs to reprogram macrophages mainly depends on PS [179,180,181]. nEVs not only act in a paracrine manner but also exert autocrine effects to reduce immune activation and promote resolution. In this context, Hurtado et al. [172] showed that nEVs modulated their own responses by delaying apoptosis and regulating inflammatory chemokine production and NET formation. A novel type of extracellular vesicles derived from aging neutrophils has been also reported by Hsu et al. [183]. They are called large aging-neutrophil-derived vesicles (LAND-Vs) as they exceed the traditional size of EVs, greater than 1 µm, and their formation is intrinsic to aging neutrophils. These LAND-Vs also exhibited anti-inflammatory activity, which was mediated by CD55 and by suppressing complement C3 activation.

The unique features of EVs such as their high stability and their ability to cross biological barriers like the blood–brain barrier make EVs an efficient tool for drug delivery, to carry a range of therapeutic agents to targeted tissues [170]. Nevertheless, due to their low yield, clinical translation becomes more complex, prompting the search for strategies to overcome these limitations. In this regard, nitrogen cavitation has been used to generate nanoscale cell membrane-formed vesicles similar to naturally secreted EVs [184,185,186,187]. These nanovesicles, loaded with resolvin D1 (RvD1) alongside the antibiotic ceftazidime, effectively reduced both pathogen growth and inflammation in models of *Pseudomonas aeruginosa*-induced lung infection [186] and peritonitis [185]. The inflammatory response was also reduced by nanovesicles loaded with the anti-inflammatory drug piceatannol in acute lung injury [187]. Similar results were observed when assessing the impact of RvD2-loaded nanovesicles in a mouse model of ischemic stroke, demonstrating a reduction in neuroinflammation through precise targeting of inflamed brain endothelium [184].

## 3. Role of Platelet-Rich Plasma as Immunomodulator of Local Neutrophils

Neutrophils are recognized as early responders to infection or injury, while platelets are the first to react to vascular damage [188,189]. Beyond their established role in hemostasis and thrombosis, platelets are increasingly recognized as key regulators of the inflammatory and immune response, primarily through dynamic interactions with immune cells, with neutrophils as their principal partners [12,13,14]. The interplay between platelets and innate immune cells not only drives thrombosis, inflammation, and tissue damage but also plays a vital role in resolving inflammation, facilitating tissue repair, and supporting wound healing [14,190,191]. It is precisely the role of platelets in wound healing, beyond hemostasis, that lies behind the use of platelet derivatives. Platelet-rich plasma (PRP) consists of a platelet concentration above the peripheral blood baseline, containing many biologically active factors including growth factors, cytokines, adhesion proteins, and plasma proteins [192,193,194]. PRP has been successfully applied in many medical fields, such as oral and maxillofacial surgery, orthopedic, ophthalmology, dermatology, gynecology, and neurology [195,196,197]. PRP has also been demonstrated to modulate the activity of several immune cells involved in the wound-healing process, such as macrophages and neutrophils [198,199]. In line with this, research exploring the immunomodulatory potential of this platelet concentrate is emerging, as its injection at the site of injury may initiate tissue repair, with platelets interacting with local immune cells [190,194] and potentially acting as immunomodulators of the neutrophil function, shifting the balance towards their reparative roles (Figure 4). However, the large number and variability of PRP protocols available result in a broad range of bioformulations with biologically heterogeneous final features (Figure 4A). This may have an impact on the clinical quality and efficacy of PRP. The main differences lie in blood collection volume, centrifugation conditions, cell composition, activation state, and application frequency and dosage [190,197,199]. Despite numerous efforts to standardize and classify PRP [200,201,202], these attempts have largely been unsuccessful. Consequently, the lack of standardization, along with the frequent absence of detailed descriptions in the literature, often leads to inconsistent and non-comparable clinical outcomes [190,198,199]. Variation in composition is one of the most controversial aspects, especially concerning whether or not to include leukocytes. The inclusion of white blood cells often leads to contamination with erythrocytes. The presence of these red blood cells results in the release of toxic hemoglobin due to hemolysis following PRP centrifugation. This, combined with the harmful pro-inflammatory effects of eryptosis, which leads to cell death, underscores the importance of avoiding their inclusion [190,203,204,205]. Additionally, the inclusion of leukocytes usually modifies the ratio of these cells in PRP with respect to the whole blood proportion, as lymphocytes are the predominant leukocytes in PRP, whereas neutrophils are the most common in blood [206]. This variation may lead to a different biological response. In addition to the aforementioned points, the inclusion of leukocytes could contribute to the exacerbation of the inflammatory response, leading to worsening pain symptoms, delayed extracellular matrix remodeling and healing, or chronic inflammation [207,208,209,210,211,212,213,214,215,216,217,218]. This might occur because the extra concentration of leukocytes added to the local population might amplify their harmful effects and overwhelm the PRP platelets’ and tissue’s ability to modulate these negative functions [209]. In fact, depletion of neutrophils in PRP increases its therapeutic efficacy [219,220]. Therefore, adding extra levels of leukocytes to the wound, rather than enhancing their reparative role, can trigger harmful pro-inflammatory effects, as a consequence of the neutrophils’ double-edged sword [221]. In contrast, pure platelet-enriched PRP could support wound healing, in part by interacting with local neutrophils and modulating their immune response (Figure 4B).

Platelet interaction with neutrophils leads to an improvement in their immune functions by increasing ROS generation, phagocytosis, and NET formation and can also aid in the resolution of inflammation by secreting pro-resolving molecules [14] and extracellular vesicles [188] (Figure 4). Platelets are capable of activating neutrophils through direct receptor-mediated physical interactions. Through this process, platelet activation leads to the exposure of P-selectin, which binds to neutrophil P-selectin glycoprotein ligand 1 (PSGL-1), triggering neutrophil activation, which in turn, induces the activation of cell surface β2 integrins on neutrophils, such as LFA-1 and Mac-1 [222,223]. The interaction between platelets and neutrophils relies not only on receptor–ligand binding but is also heavily influenced by soluble factors [223] (Figure 4). In fact, platelets also release the representative DAMP high mobility group box 1 (HMGB1) protein, chemokines such as CXCL4, CCL5, and CXCL7, and microparticles that are actively involved in neutrophil recruitment and activation [222,223,224,225]. This platelet–neutrophil complex acts bidirectionally, as neutrophil activation also amplifies platelet activation [222]. As previously mentioned, platelets modify a number of neutrophil reparative functions. Neutrophil phagocytosis of periodontopathogens was enhanced by platelets through direct interaction, requiring functional toll-like receptor 2 (TLR2) [226]. Additionally, Miyabe et al. [227] reported that PRP enhanced neutrophilic phagocytosis via the activation of iC3b receptors (CD11b/CD18, Mac-1). Under severe conditions like sepsis, platelets strongly activate neutrophils and drive NET formation through TLR4, thereby increasing neutrophils’ ability to trap bacteria [228]. In contrast, Rui et al. [229] reported that PRP exosomal miR-26b-5p improved diabetic wound healing by inhibiting NETs through the direct suppression of its target gene *MMP-8*, whose overexpression is involved in non-healing chronic ulcers. NET formation was associated not only with pathogens but also with cytokines, chemokines, and platelet agonists [12]. NETs also trigger platelet activation, a process that relied on integrin αIIbβ3 [230], which is a key adhesive molecule for platelets binding endothelium. These immobilized platelets provided an essential foundation for neutrophil recruitment, facilitating tissue repair in a sterile liver injury model where NET release was limited [231]. Indirect interaction mediated by pro-resolving mediators also plays an important role. In this regard, maresin 1 production was dependent on platelet–neutrophil interactions and reported to be organ-protective in a murine model of acute respiratory distress syndrome [232].

Therefore, the immunomodulatory interactions between platelets and neutrophils provide a promising basis for developing new therapeutic strategies. These interactions represent one of the mechanisms through which PRP can promote wound healing, a field that remains largely unexplored.

## 4. Concluding Remarks

Neutrophils, as part of the innate immune system, are the initial responders at injury sites. They are classically considered as the frontline defenders, possessing potent microbicidal activity. Their arsenal of cytotoxic mechanisms and inflammatory mediators, however, can also cause collateral tissue damage. Nevertheless, over the last decade, neutrophils have increasingly been recognized as effectors of inflammation resolution and tissue repair (Table 1). The significant phenotypic diversity and functional adaptability of neutrophils contribute to their remarkable duality. This versatility allows neutrophils to play a crucial role in defense, simultaneously affecting tissue damage and chronic inflammation, as well as promoting wound healing and resolution. We therefore aimed to focus on the mechanisms and strategies involving neutrophils in the context of wound healing and tissue repair. Neutrophil contributions to healing are clearly context-dependent, influenced by numerous factors. They exhibit several active programs that set in motion the resolution of inflammation. These programs include clearing necrotic cellular debris, releasing several pro-resolving mediators and extracellular vesicles, NET formation in response to a range of stimuli, neutrophil migration away from the source of inflammation, and neutrophil apoptosis that promotes efferocytosis and macrophage polarization. In this regard, neutrophil-dependent macrophage reprogramming is a mechanism that occurs across all these processes, enabling neutrophils to indirectly modulate macrophage function, eliciting an anti-inflammatory and pro-resolution response. Additionally, neutrophils’ circadian rhythms are involved in regulating this immune response, whereby neutrophils autonomously express biological clock-regulating genes. Modulating these specific neutrophil activities may offer a potential target for the development of new therapeutic targets. Owing to the dual role of neutrophils, their global inhibition is not recommended, given that depletion has led to adverse effects including impaired wound healing [233,234] and worsened colitis [235,236]. Therefore, controlling neutrophil resolution and their immunomodulation is one of the major clinical challenges.

In the context of tissue repair, PRPs are essential due to the significant role of platelets in wound healing. Platelets, as key regulators of the inflammatory and immune response, interact with neutrophils to modulate their function, thereby enhancing their reparative capabilities. PRP may, therefore, contribute to enhancing neutrophil recruitment, activation, and activity, thereby promoting tissue repair. However, the wide range of PRP protocols and the lack of standardization result in biologically diverse formulations, leading to varying clinical outcomes. In this context, the inclusion of leukocytes is particularly contentious, as many studies have indicated detrimental effects when these white blood cells are added to blood-derived products. Additionally, plasma proteins may also interact with neutrophils, reducing their activity, and exhibited anti-inflammatory effects, as reported by Ngoc et al. [215], when using platelet-poor plasma. Just as neutrophil biology encompasses a spectrum of functional states beyond simplistic dichotomies, their interactions with platelets similarly reflect an intricate and dynamic interplay. Understanding the complex crosstalk between pure platelet-enriched PRP and local neutrophils might reveal new therapeutic opportunities for blood-derived products to mediate wound healing and tissue repair.

In summary, neutrophils have emerged as a promising therapeutic approach due to their remarkable phenotypic plasticity and their crucial role in resolving inflammation and restoring homeostasis. However, their activity must be finely regulated to prevent potential tissue damage. The circadian regulation of innate neutrophil activation could have significant implications for creating personalized therapies tailored to specific diseases. Further advances in neutrophil-targeted therapies will allow for precise control over the platelet–neutrophil interaction to dampen inflammation and favor reparative processes. In this context, PRPs have significant contributions to make. Further research is needed to unravel the complexity of neutrophil biology, enabling the successful translation of new insights into clinical practice.

## Figures and Tables

**Figure 1 ijms-26-08669-f001:**
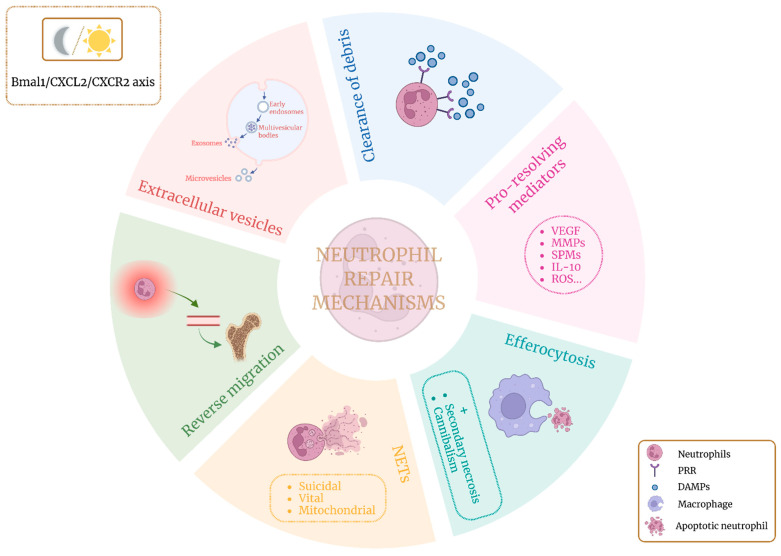
Overview of the main mechanisms by which neutrophils contribute to tissue repair. These processes are also regulated by circadian rhythms, as neutrophils autonomously express biological clock-regulating genes. Created with BioRender.com.

**Figure 2 ijms-26-08669-f002:**
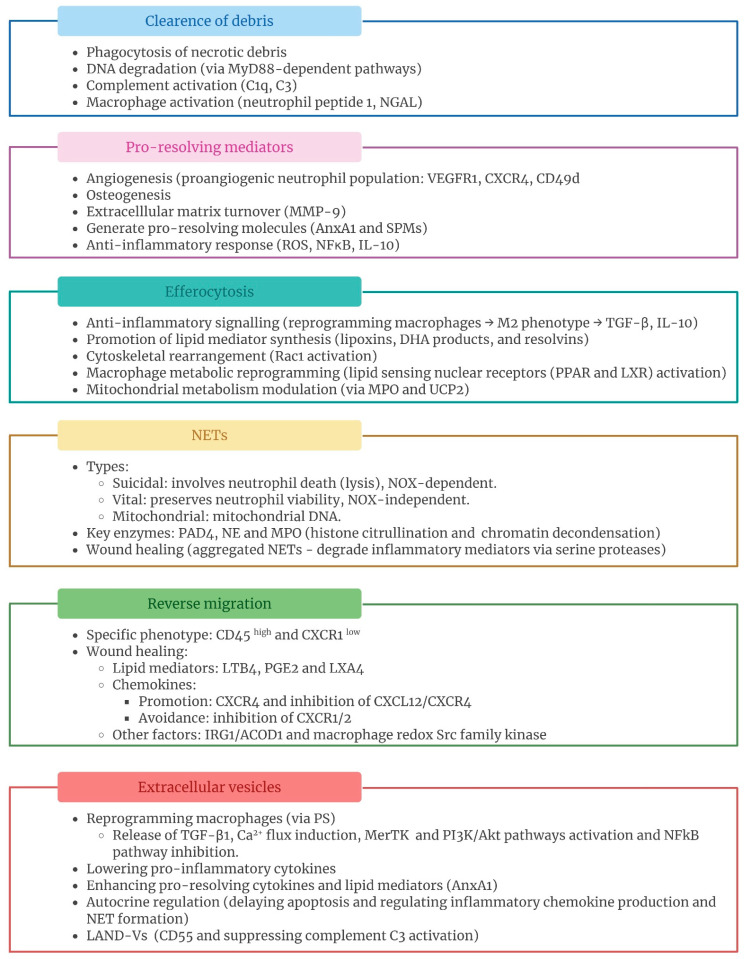
Summary of the principal features and functions of the processes through which neutrophils contribute to tissue regeneration. Each process is outlined with its principal features and the mechanisms by which it exerts its biological function. Created with BioRender.com. AnxA1: annexin A1; CD49d: integrin alpha 4; CXCL12: C-X-C motif chemokine ligand 12; CXCR1/2: C-X-C chemokine receptors 1 and 2; CXCR4: C-X-C chemokine receptor type 4; DHA: docosahexaenoic acid; IL-10: interleukin-10; IRG1/ACOD1: immune-responsive gene 1/aconitate decarboxylase 1; LAND-Vs: large aging-neutrophil-derived vesicles; LTB4: leukotriene B4; LXA4: lipoxin A4; LXR: liver X receptor; MMP9: matrix metalloproteinase-9; MPO: myeloperoxidase; NE: neutrophil elastase; NETs: neutrophil extracellular traps; NF-κB: nuclear factor kappa-light-chain-enhancer of activated B cells; NGAL: neutrophil gelatinase-associated lipocalin; NOX: NADPH oxidase; PAD4: peptidylarginine deiminase 4; PGE2: prostaglandin E2; PPAR: peroxisome proliferator-activated receptor; PS: phosphatidylserine; ROS: reactive oxygen species; SPMs: specialized pro-resolving mediators; TGFβ: transforming growth factor beta; UCP2: uncoupling protein 2; VEGFR1: vascular endothelial growth factor receptor 1.

**Figure 3 ijms-26-08669-f003:**
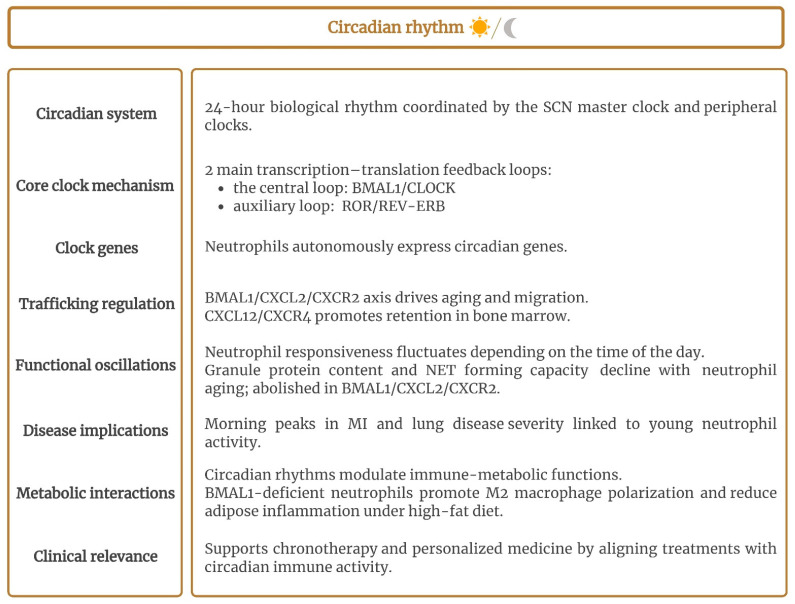
Summary of how circadian rhythm influences neutrophil characteristics and functions, emphasizing the role of the circadian system and core clock genes in regulating neutrophil trafficking, functional rhythms, and metabolic interactions. The figure also outlines the clinical implications of these temporal dynamics, including their relevance to disease development and immune regulation. Created with BioRender.com. BMAL1/CLOCK: brain and muscle ARNT-like 1/circadian locomotor output cycles kaput; BMAL1/CXCL2/CXCR2: BMAL1/C-X-C motif chemokine ligand 2/C-X-C chemokine receptor 2; CXCL12/CXCR4: C-X-C motif chemokine ligand 12/C-X-C chemokine receptor 4; MI: myocardial infarction; NET: neutrophil extracellular trap; ROR/REV-ERB: retinoic acid receptor-related orphan receptor/reverse-Erb nuclear receptor; SCN: suprachiasmatic nucleus.

**Figure 4 ijms-26-08669-f004:**
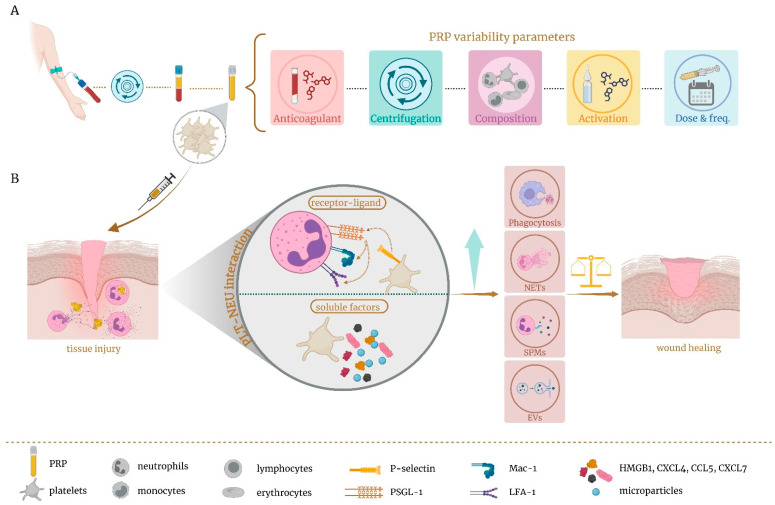
Neutrophils and platelet-rich plasma. (**A**) PRP protocols vary widely in their parameters, which can significantly influence their overall effectiveness. (**B**) Immunomodulatory role of platelets. When focusing on platelet-rich plasma formulations enriched exclusively with platelets, their application to injured tissue can trigger interactions between the platelets and neutrophils present at the wound site. These interactions may occur directly through receptor–ligand binding or indirectly via soluble agonists. Both pathways contribute to neutrophil activation and enhance their reparative functions, ultimately supporting tissue regeneration. Created with BioRender.com. EVs: extracellular vesicles; HMGB1: high-mobility group box 1; Plt-Neu: platelet–neutrophil; PRP: platelet-rich plasma; PSGL-1: P-selectin glycoprotein ligand 1; SPMs: specialized pro-resolving mediators.

**Table 1 ijms-26-08669-t001:** Overview of neutrophil-mediated tissue repair across organs. The table presents in vivo studies focused on specific diseases or targeted organs. Studies investigating general inflammatory mechanisms, without reference to a particular disease, are not included in the table but are discussed in the main text. aggNET: aggregated neutrophil extracellular traps; AMPK: AMP-activated protein kinase; AnxA1: annexin 1; BLT1: leukotriene B4 receptor 1; BMAL1brain and muscle ARNT-like 1; CCN1: communication network factor 1; CXCR: CXC chemokine receptors; EGF: epidermal growth factor; EVs: extracellular vesicles; GJA1: gap junction alpha-1 protein; HMGB1: high-mobility group box 1; IL-10: interleukin-10; IRG1/ACOD1: immune-responsive gene 1/aconitate decarboxylase 1; JAM: junctional adhesion molecule; LAND-Vs: large aging-neutrophil-derived vesicles; miR: microRNA; MMP: matrix metalloproteinase; NETs: neutrophil extracellular traps; PPARγ: peroxisome proliferator-activated receptor gamma; PS: phosphatidylserine; ROS: reactive oxygen species; RvD: resolvin D; SOD2: superoxide dismutase 2; VGEF: vascular endothelial growth factor.

Organ/Tissue	Mechanism	Specific Strategy	Reference
Adipose	Circadian rhythm	*Bmal1*	[163]
Bone	EVs	SOD2 and GJA1	[171]
Brain	Circadian rhythm	NETs	[166]
Brain	EVs	Drug delivery (RvD2)	[184]
Eye	NETs	aggNET	[115]
Heart	Circadian rhythm	CXCR2^high^	[168]
	Clearance of debris	Macrophage activation	[31]
	Pro-resolving mediators	MMP-9	[41]
	Reverse migration	CXCR1/2	[145]
Ischemic tissue	Pro-resolving mediators	MMP-2, MMP-9	[44]
		VEGF, MMP-9	[38,45]
Joints	Efferocytosis	Cannibalism	[97]
	EVs	(miR)-455-3p	[177]
	EVs	AnxA1	[178]
	EVs	PS and AnxA1	[181]
	NETs	aggNET	[114]
	Pro-resolving mediators	Resolvins	[55]
Kidney	Circadian rhythm	CXCR2+ neutrophils	[154]
	Reverse migration	Reverse migration	[129]
Liver	Clearance of debris	Complement activation	[19]
		DNA degradation	[23]
	Pro-resolving mediators	MMPs	[46,47]
		ROS	[60]
	Reverse migration	CXCR4	[23]
Lung	Circadian rhythm	Neutrophil proteome	[165]
	Clearance of debris	DNA degradation	[22]
	Efferocytosis	PPARγ	[85]
		Cannibalism	[95]
		Active migration and EGF	[86]
		Neutralization HMGB1	[87]
		HMGB1 (inhibitory effects)	[90]
		HMGB1(inhibitory effects)	[92]
		Mitochondrial AMPK axis	[93]
		Mitochondrial metabolism modulation	[84]
	EVs	Drug delivery (piceatannol)	[187]
		Drug delivery (RvD1 and ceftazidime)	[186]
		LAND-Vs	[183]
	Pro-resolving mediators	IL-10	[65]
	Reverse migration	IRG1/ACOD1	[147]
		JAM-C	[139]
Muscle	Clearance of debris	Clearance of debris	[30]
Nervous system	Clearance of debris	Clearance of myelin debris	[25,27]
		Macrophage activation	[28,29]
Pancreas	NETs	aggNET	[123]
	Pro-resolving mediators	VEGF, MMP-9	[43]
	Reverse migration	BLT1	[138]
Periodontium	NETs	aggNET	[116]
Peritoneum	EVs	Drug delivery (RvD1 and ceftazidime)	[185]
Rectum	NETs	NET-associated immunothrombi	[120]
Skin	Efferocytosis	CCN1	[68]

## Data Availability

This manuscript is a review article; therefore, it does not involve the use of any original research data.

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
