# Peer review of "Beyond Killing: The Overlooked Contribution of Neutrophils to Tissue Repair"

_ijms, 2025, doi:10.3390/ijms26178669_

Round 1
Reviewer 1 Report
Comments and Suggestions for Authors
This review article presents a comprehensive and timely synthesis on the emerging roles of neutrophils in tissue repair, extending beyond their classical cytotoxic and inflammatory functions. The manuscript is rich in content, well-structured, and supported by a broad and up-to-date bibliography. The authors clearly articulate the multifaceted role of neutrophils in immune resolution and regeneration, including mechanisms such as efferocytosis, NET formation, extracellular vesicle release, and circadian modulation. Moreover, the discussion on the interplay between neutrophils and platelet-rich plasma (PRP) is particularly novel and may have clinical relevance.
However, despite the manuscript’s merits, there are several areas that require clarification, reorganization, and improved scientific precision. Below are specific comments and suggestions for improvement.
Major Comments
- The manuscript provides a wide-ranging overview of neutrophil functions but lacks a clearly defined central hypothesis or guiding framework that connects the sections. I suggest the authors emphasize a unifying conceptual model at the beginning and reiterate it in the concluding remarks.
- While the section on PRP and its interaction with neutrophils is interesting, it occupies a significant portion of the manuscript and somewhat shifts the focus away from the primary topic of neutrophil biology. Unless the PRP angle is a core thesis of the manuscript, I recommend condensing this section or integrating it more tightly with the discussion on neutrophil reparative functions.
- Certain mechanistic discussions, particularly those involving MMP-9, VEGF, and efferocytosis, are repeated across different sections. A consolidation of these themes would improve flow and readability.
- The figures are visually appealing but need to be better integrated into the text. The authors should clearly reference the specific figure panels when discussing mechanisms in the main text. In addition, a higher resolution or a supplementary table summarizing key neutrophil reparative mechanisms may aid comprehension.
- The authors disclose their affiliation with BTI Biotechnology Institute and its work on PRGF-Endoret technology. While this is appreciated, given the extensive focus on PRP, it would be appropriate to provide a more neutral tone when discussing its clinical potential and to cite independent studies as well.
Minor Comments
- The manuscript is generally well-written but would benefit from a thorough language revision for conciseness and clarity. There are occasional awkward phrasings, such as “just as the world of neutrophils isn’t simply black and white...” which may be too colloquial for a scientific review.
- Please ensure all abbreviations are defined upon first use (e.g., LAP, TIMP, GAS6, etc.). Also consider including a list of abbreviations at the end for ease of reference.
- Some references are cited multiple times in close proximity (e.g., references on MMPs and efferocytosis). The authors might consolidate these or ensure unique contributions are clearly stated.
Reviewer 2 Report
Comments and Suggestions for Authors
This is an exhaustively researched and wide-ranging review that presents a new and timely view of the complex roles of neutrophils in tissue repair beyond their classically recognized microbicidal role. The manuscript is written clearly and is heavily referenced, with good conceptual integration and logical flow. The fact that platelet-rich plasma (PRP) modulation of neutrophil activity is included makes the manuscript highly relevant to translational and clinical applications. The manuscript, however, needs to be improved structurally, with better integration of figures, concise expression in certain sections, and better demarcation between original data and literature review.
1. The present title is descriptive but can be more compact. Suggested: "The Reparative Role of Neutrophils: Beyond Inflammation and Microbicidal Activity."
2. The paper tries to describe numerous mechanisms (e.g., NETs, EVs, circadian rhythm, PRP interaction), but the text gets too exhaustive. Try grouping mechanisms into thematic clusters to make it easier to read.
3. The abstract is thorough, but the mention of circadian rhythms might come across as sudden. Just state briefly some key examples or implications (e.g., treatment timing or immune activity variation).
4. Figure 1 and Figure 2 are useful but too briefly referred to in the main text. Try to expand on the main message of each figure when you refer to them.
5. Several paragraphs repeat similar statements on neutrophil duality. Editing for brevity and eliminating redundant phrasing will enhance impact.
6. The PRP and platelet section (especially Section 13 onward) is inordinately lengthy in proportion to its novelty. Some of this material could be cut or relegated to a table/figure.
7. Although the review is extensive and thorough, it would be improved by an additional paragraph that addresses translational gaps, open questions, or potential future therapeutic avenues for neutrophil plasticity.
8. The circadian section reads well but could potentially confuse general readers. Think about more briefly summarizing the molecular loops and identifying direct neutrophil relevance.
9. The abbreviation "aggNETs" is used without being defined in the main text. Perhaps define briefly at first use.
10. The paper addresses PRP inconsistencies nicely, but could be improved with a table comparing important differences between PRP formulations and their recognized immunological effects.
11. Certain parts (e.g., on EVs and efferocytosis) are citation-heavy. The references could be condensed or recent reviews cited to make the flow more streamlined.
12. A diagram outlining neutrophil reparative roles in different tissue types (e.g., muscle, CNS, bone, skin) would increase the reader's comprehension.
13. Although the authors report affiliations with a PRP-developing institution, more declarative statements delineating scientific interpretation from commercial interest would enhance trust.
14. The conclusion is complete but would be improved with a more action-oriented tone (e.g., "We suggest that.", "Future research needs to address.").
15. Provide uniform formatting for abbreviations (e.g., PRP, NETs, EVs) and citations throughout.
Comments on the Quality of English LanguageThe manuscript is mostly well-written and effectively communicates intricate biological ideas. Nevertheless, certain sections would gain from linguistic refinement to enhance clarity, eliminate redundancy, and increase professional polish.
Round 2
Reviewer 1 Report
Comments and Suggestions for Authors
The authors have adequately addressed all of my previous comments. I would only suggest correcting the format of the references in accordance with the journal’s guidelines and placing the title of the last table above it. Once these minor revisions are made, the manuscript can be accepted.
Reviewer 2 Report
Comments and Suggestions for Authors
Most of the previous round reviewer comments have been satisfactorily addressed by the authors, with good responses and suitable changes in content and presentation.
In particular, the addition of more figures and a table enhances the readability and ease of comprehension of intricate mechanisms elaborated. The abstract has been modified to include circadian rhythm relevance more directly, and some editorial enhancements are incorporated throughout the manuscript. Although some suggestions were noted but not completely acted on (e.g., condensing the PRP section), the authors gave reasonable explanations substantiated by the scope and translational relevance of the manuscript.
In general, the revised version is considerably better in structure, clarity, and depth. I suggest acceptance with minor revisions as specified below.
1. While a list of abbreviations has been included, it is necessary to define all abbreviations (e.g., PRP, NETs, EVs) at their first use in the main text to facilitate understanding for readers who are non-experts.
2. Think about briefly augmenting figure legends, particularly for Figures 2 and 3, to make clear the main points without needing back-reference to the text.
3. There are some formatting problems remaining (e.g., unnecessary line breaks, lack of consistency in use of italics for gene/protein names). Final proofreading is suggested.
4. While the focus on neutrophil duality is deliberate, some paragraphs would be more effective with judicious trimming to enhance conciseness without loss of emphasis.
